# Effects of Tenofovir Disoproxil Fumarate on Bone Quality beyond Bone Density—A Scoping Review of the Literature

**DOI:** 10.3390/ph17020146

**Published:** 2024-01-23

**Authors:** Tejpal Singh Hashwin Singh, Tejpal Singh Jashwin Singh, Kok-Yong Chin

**Affiliations:** Department of Pharmacology, Faculty of Medicine, Universiti Kebangsaan Malaysia, Cheras 56000, Malaysia; hashwinsingh2000@gmail.com (T.S.H.S.); jashwinsingh2004@gmail.com (T.S.J.S.)

**Keywords:** antiretroviral therapy, bone biomechanical strength, bone microstructure, osteoporosis, osteopenia, skeleton

## Abstract

Tenofovir disoproxil fumarate (TDF) is a widely used pharmacological agent for the treatment of human immunodeficiency virus infection. While prolonged exposure to TDF has been associated with a decrease in bone mineral density (BMD) and increased fracture risk, limited discussion exists on its effects on various aspects of bone quality. This scoping review aims to provide a comprehensive overview of the impact of TDF on bone quality beyond BMD. A literature search was conducted using the PubMed and Scopus databases to identify studies investigating the effects of TDF on bone quality. Original research articles written in English, irrespective of study type or publication year, were included in the review. Seven articles met the inclusion criteria. Findings indicate that prolonged exposure to TDF adversely affects bone microarchitecture and strength, impeding fracture healing and skeletal microdamage repair. The observed effects suggest a complex interplay involving bone cell signalling, cytokines and bone remodelling processes as potential mechanisms underlying TDF’s impact on bone quality. As a conclusion, TDF impairs bone remodelling and microarchitecture by influencing dynamic bone cell behaviour and signalling pathways. Future studies should delve deeper into understanding the intricate negative effects of TDF on bone and explore strategies for reversing these effects.

## 1. Introduction

The human immunodeficiency virus (HIV), belonging to the Lentivirus genus within the Retroviridae family, is primarily recognised for its consequential immunologic effects in humans due to its replication in CD4+ T lymphocytes and monocytes/macrophages, ultimately resulting in severe lymphopenia and immunodeficiency if left untreated. The fatal effects of this virus, if left untreated, are not merely restricted to the cellular level but are widespread throughout various bodily organs and systems such as the heart, kidney, lungs, liver and musculoskeletal and central nervous system [1]. The Global Burden of Disease report indicated that there were 36.85 million cases of HIV/AIDS in 2019, which shows an increment of 307.26/100,000 cases compared with 1990. Similarly, there were 863.84 thousand deaths in 2019 due to HIV/AIDS, which showed an increase of 10.72 deaths/100,000 cases compared with 1990 [2]. 

Given the devastating effects of HIV, a therapy regimen known as combination antiretroviral therapy was introduced in the mid-1990s. It currently boasts more than thirty drugs that have been approved for the treatment of HIV-positive individuals [3]. The current combination of antiretroviral drugs can be further divided into five main classes that target distinct steps in the HIV cycle. These classes include HIV entry inhibitors such as enfuviritide/ENF (Fuzeon), nucleoside reverse transcriptase inhibitors (NRTIs) such as tenofovir disoproxil fumarate/TDF (Viread), non-nucleoside reverse transcription inhibitors (NNRTIs) such as efavirenz/EFV (Sustiva), integrase strand transfer inhibitors (INSTIs) such as raltegrivir (Isentress) and protease inhibitors such as ritonavir (Norvir) [4]. Despite the wide array of drugs available for the treatment of HIV, current guidelines strongly support the use of a combination regimen of INSTIs and NRTIs as initial therapy for most people with HIV [5]. 

TDF is one of the most widely used antiretroviral agents in the world [6]. TDF is a NRTI that competitively inhibits HIV reverse transcriptase, which consequently prevents the transcription of viral ribonucleic acid (RNA) to viral deoxyribonucleic acid (DNA). This then halts a cascade of reactions which include the following, in order: integration of viral DNA into the host’s DNA following translocation of viral DNA to the nucleus, transcription of mRNA coding for viral proteins, translation to proteins and post-translational cleavage by HIV protease and lastly viral maturation, budding and replication [6]. The widespread use of TDF can be attributed to its extended half-life in plasma and intracellularly, its low risk of developing viral resistance mutations and its favourable tolerability by patients, which improves patients’ compliance [7]. Apart from that, tenofovir (or entecavir) is also indicated for patients with hepatitis B virus infection, which substantially increases the number of users of this drug [8]. 

Despite the impressive efficacy and tolerability of TDF, there are a plethora of undesirable side effects that are associated with its use. TDF has been shown to induce cellular toxicity and damage to the proximal tubule of the kidney, which, if left untreated and progresses, ultimately results in the decline of kidney function, osteomalacia and pathological fractures [6,9,10]. The use of TDF has also been positively associated with low bone mineral density (BMD) [11,12,13]. Individuals exposed only to TDF exhibited a more prominent loss in BMD as compared with those who were exposed to other antiretroviral drugs or a multi-drug regimen [11,12,13,14,15,16,17]. A randomised trial of TDF and emtricitabine as pre-exposure prophylaxis (PrEP) in 498 HIV-seronegative men who have sex with men (MSM) and transgender women from Thailand, South and North America showed a significant decrease in spine and hip BMD amongst participants randomised to emtricitabine/TDF PrEP after 24 weeks of treatment [18]. A significant decrease in lumbar spine, total hip and total body BMD Z-scores was observed in another clinical trial in the United States involving 101 MSM (15–22 years old), whereby participants with the highest exposure to TDF exhibited the greatest decline in total hip and femoral neck BMD [19]. Similarly, in patients with chronic hepatitis B infection, treatment with TDF was associated with lower BMD compared with controls and a more rapid decline in TDF compared with patients on entecavir [20,21]. In addition, switching patients with chronic hepatitis B infection and normal baseline BMD from TDF to entecavir or tenofovir alafenamide improved their BMD after 96 weeks [22]. These observations are especially worrying for older adults on TDF because they are already suffering from osteoporosis or are at high risk for fractures [23].

Although there is extensive research and evidence available regarding the detrimental effects of TDF use on BMD, its effects on other aspects of bone quality remain elusive. It is not possible to develop an effective agent or regimen to combat the destructive effects of TDF on bone if the mechanisms that cause these effects are unknown. Therefore, this paper aims to analyse findings from studies on these mechanisms to further understand the mechanism of bone loss induced by TDF. Ultimately, this paper hopes to propel the efforts in developing an agent that may reduce the destructive effects of TDF on bone and amplify the understanding of the scientific community on the mechanisms of TDF-induced bone loss to further enhance prescribing guidelines of TDF to maintain the quality of life of patients.

## 2. Results

The literature search, encompassing the PubMed and Scopus databases, identified 2442 unique articles, with 481 duplicates removed. Subsequent screening of titles and abstracts led to the exclusion of 2435 articles due to various reasons, including not being original research articles (*n* = 113), being conference abstracts (*n* = 15) or falling outside the scope of the current review (*n* = 2307). The full texts of the remaining seven articles were then retrieved and analysed, meeting the criteria for inclusion in the review (Figure 1).

Aligning with the objective of examining bone quality and understanding the mechanisms of bone loss induced by TDF or in combination with other drugs, the seven selected studies employed diverse animal models: rhesus monkeys (*n* = 1), rats (*n* = 2), mice (*n* = 2) and zebrafish (*n* = 1). Additionally, one study focused on patients taking TDF. Notably, one investigation explored the impact of antiretroviral therapy (ART), including TDF, on fracture healing, while the others administered TDF to intact animals. The assessment of bone quality and remodelling processes was predominantly conducted through techniques such as skeletal micro-computed tomography (µCT), bone histomorphometry, immunohistology, bone biomechanical strength and bone remodelling markers, providing a comprehensive understanding of TDF-induced effects on skeletal health.

Carnovali et al. (2016) explored the effects of TDF on zebrafish osteogenesis and scale remodelling. While TDF did not significantly impact the mineralisation rate in embryos, it modulated bone metabolism markers in adult fish, blocking scale growth and increasing the resorption area. This implies potential catabolic effects that span developmental stages [24].

Conesa-Buendía et al. (2019) conducted in vitro and in vivo studies involving bone marrow cells and mice treated with TDF. TDF exhibited complex effects, inhibiting osteoblast differentiation and increasing osteoclast activity in vitro and in vivo. The effects of TDF on osteoclasts could be mediated by enhanced nuclear factor kappa B (NFκB) signalling and pannexin-1 or connexin-43 function, in conjunction with increased levels of receptor activator of nuclear factor kappa beta (RANKL) and decreased osteoprotegerin (OPG) by osteoblasts. TDF also decreased cortical and trabecular bone volume and trabecular number and increased trabecular separation in mice, coincidentally with the decreased mineral apposition rate. The effects of TDF on osteoblasts could be mediated by the decreased nuclear translocation of β-catenin. These effects translated to decreased BMD and altered bone markers in TDF-treated mice, emphasising the need for a comprehensive understanding of its mechanisms [25].

In a study by Conradie et al. (2017), 12- to 14-week-old male Wistar rats were orally treated with TDF, stavudine or a control for 9 weeks. While TDF did not show significant differences in BMD compared with the control group, it reduced bending stress and mineralising surface area, indicating potential effects on bone mechanical strength and mineralisation activity. Although bone cellular histomorphometry was not impacted, lipid droplets in bone marrow increased significantly with TDF treatment. Compared with stavudine, the negative impact of TDF on the skeleton was less severe [26]. 

Matuszewska et al. (2020) investigated 8-week-old male albino Wistar rats treated with TDF, efavirenz, or a control for 24 weeks. The TDF group displayed reductions in femoral indices, femoral weight, mid-femoral diameter and total BMD after 24 weeks. Microscopically, the femoral trabecular number increased and trabecular separation decreased in the TDF group. Biomechanically, Young’s modulus also decreased in the TDF group. These findings suggest potential adverse effects on bone integrity and strength with prolonged TDF exposure, highlighting the importance of considering the duration of treatment. However, the bone remodelling markers did not change with TDF treatment [27]. 

Castillo et al. (2002) investigated growing rhesus monkeys treated with TDF and infected with simian immunodeficiency virus (SIV). TDF increased tibial osteoid seam width, a marker of bone microdamage, regardless of SIV infection. Additionally, SIV infection itself increased resorption cavity density, indicating potential combined effects on bone health that warrant further investigation [28].

Graham et al. (2022) explored the effects of ART containing TDF, lamivudine and efavirenz on fracture healing in Wistar rats. While an initial decrease in fracture healing union rate was observed at week 4, no significant differences in biomechanical strength were noted between the ART and control groups at week 8. This underscores the importance of studying the temporal dynamics of bone healing under ART regimens [29].

Ramalho et al. (2019) conducted a study on ART-naive male individuals with HIV initiated on TDF-based therapy. Post-treatment, there was a decrease in BMD at various sites, alterations in bone turnover markers and cytokine expression, improved cortical thickness and changes in osteoblast and osteoclast parameters. These multifaceted effects highlight the complexity of TDF’s impact on bone health [30].

The included studies are summarised in Table 1. In summary, the studies collectively reveal the diverse effects of TDF on bone health across species and models, impacting various aspects, such as structure, mechanical strength and turnover markers. The findings underscore the importance of considering factors like treatment duration and the temporal dynamics of bone healing in understanding TDF-related bone effects and their potential clinical implications.

## 3. Discussion

Bones are perpetually remodelling tissues that serve essential roles in sustaining human life such as providing mechanical support, permitting locomotion, protecting vital organs and being the main haematopoietic organ for adults. Therefore, the continuous remodelling of bone to preserve its strength and avoid the accumulation of damage is vital for optimal functioning [31,32]. Bone remodelling is essentially a process where osteoclasts resorb old or damaged bone and osteoblasts form new bone [33]. In this review, the effects of TDF on various aspects of bone quality through modulation of bone remodelling are discussed.

### 3.1. Effects on Osteoblasts

Osteoblasts derive from mesenchymal stem cells and mineralised the skeleton [33,34,35]. TDF was shown to induce a significant age- and dose-dependent decrease in ALP, an osteoblastic marker, in a study conducted by Carnovali et al. using a zebrafish model [24]. TDF treatment was also shown to significantly increase the bone marrow adiposity in the femurs of rats, which indicates the potential ability of TDF to skew progenitor cell differentiation away from osteoblastogenesis towards adipogenesis, thus decreasing osteoblasts [26,36]. A histomorphometry analysis in a study conducted by Conesa-Buendía et al., in which a mouse model was used, also showed a significant decrease in the number of osteoblast cells per surface. Furthermore, this study also indicated that TDF treatment inhibited osteoblast differentiation in a dose-dependent manner [25]. These data suggest that TDF treatment results in a decrease in osteoblast activity, which tallies with a previous in vitro study that demonstrated an alteration in the expression of genes involved in various vital cellular signalling pathways and in amino acid biosynthesis and metabolism that ultimately inhibited osteoblast differentiation, growth and activity in primary osteoblasts that were treated with TDF [37]. 

### 3.2. Effects on Osteoclasts

Osteoclasts originated from myeloid/monocyte precursors are the cells primarily responsible for bone resorption. Their differentiation is stimulated by RANKL and suppressed by OPG secreted by osteoblasts [38,39,40,41,42,43]. TDF treatment results in increased activity of osteoclasts, as evidenced by the significant elevation in the expression of TRAP, an osteoclastic activity marker, in various studies [24,25]. A study conducted by Conesa-Buendía et al. demonstrated the various mechanisms through which TDF treatment significantly enhanced osteoclast differentiation. In this study, TDF treatment significantly increased the following: RANKL mRNA expression, NFATc1 expression, ERK1/2 phosphorylation, ERK1/2, p38 and NFkB nuclear translocation and RAW264.7 macrophage differentiation. The enhancement of these gene expressions and cellular signalling pathways ultimately resulted in an increased rate of osteoclast differentiation [25]. Various studies have highlighted the ability of TDF to cause a significant elevation in CD4+ and CD68+ macrophages, which serve as precursor cells to osteoclasts [25,30]. TDF also increased the expression of cathepsin K, a lytic enzyme secreted by osteoclasts during the degradation of bone [25]. OPG mRNA expression was also decreased by TDF treatment, which resulted in a net increase in osteoclast differentiation and activation that increases bone resorption [25]. 

### 3.3. Effects on Bone Remodelling and Quality

A tightly regulated balance between osteoblastic bone formation and osteoclastic bone resorption exists in normal bone physiology. This ensures the establishment of a homeostatic environment that prevents major alteration in net bone mass and mechanical strength. Alternation in bone remodelling skewing towards the bone resorption process will result in net bone loss and osteoporosis [44]. Two major types of bone exist, i.e., cortical bone, which provides mechanical strength and protection, and trabecular bone, in which most of the metabolic functions of the bone occur. The trabecular bone is therefore primarily affected by pathologies in bone remodelling [32]. An imbalance between bone formation and bone resorption will result in abnormal bone remodelling, which may result in the development of an osteoporotic phenotype which is characterised by low bone mass, structural deterioration of bone, increased bone fragility and increased vulnerability to fractures [45].

TDF treatment results in a dysregulation of physiological bone remodelling by promoting osteoclast function while simultaneously depressing osteoblast activity, which results in a net increase in bone resorption and bone loss. This was made further evident by a study by Conradie et al. which showed that TDF treatment significantly lowered the percentage of mineralising surfaces and resulted in a slight tendency towards a lower bone formation rate in the femurs of rats [26]. Moreover, bone loss was also observed in another study utilising Wistar rats where all the rats receiving TDF treatment showed significantly lower total body BMD, femoral indices, femoral weights, mid femoral and tibial diameters, bone surface/tissue volume and number of trabeculae when compared with the control group [27]. When compared with the control, the femurs of the rats treated with TDF in this study also exhibited significantly increased trabecular separation, which is indicative of an osteoporotic phenotype and consequently a lower biomechanical strength and increased risk of fracture, as demonstrated by a significantly lower Young’s modulus [46]. The scales of old zebrafish were not spared from an osteoporotic phenotype during TDF treatment wherein TDF completely blocked the physiological scale formation of 6-month and 12-month-old zebrafish [24]. The effects of TDF treatment on individuals with developing bones are also dire; a study by Castillo et al. showed that growing rhesus monkeys treated with TDF showed a significantly increased osteoid seam width as compared with the control group [28]. Osteoid is unmineralised and softer than bone which has been mineralised. An increased amount of osteoid may result in microdamage to the bone and ultimately bone fragility [47]. This evidence suggests that TDF treatment may result in the alteration of the mineralisation of bone, termed osteomalacia [48]. This puts younger patients on TDF treatment at risk of orthopaedic complications such as kyphoscoliosis and insufficiency fractures known as looser zones in the femoral neck, pubic and ischial rami [49,50]. It is also important to note that TDF treatment delays the process of bone healing due to its mechanism of inhibiting osteoblast-mediated bone formation and promoting osteoclast-mediated bone resorption, as evidenced by a study by Graham et.al, where the fractured tibias of Wistar rats treated with TDF, lamivudine and efavirenz for four weeks showed a significantly lower union rate as compared with the rats in the control group. These ununited fracture sites also revealed a clear gap at the fracture ends filled with fibrous tissue where no woven bone nor cartilaginous callus was found in the inter-fragmentary area, which further highlights TDF’s ability to suppress osteoblast-mediated bone formation [29].

The effects of TDF on the bone remodelling process and various aspects of bone quality are summarised in Figure 2.

### 3.4. Limitations

The current research field is not without limitations. As individuals with HIV live longer with proper treatment, prolonged exposure to TDF may elevate their risk of osteoporosis in their later years. This concern is particularly relevant in the context of age-related hormone deficiencies resulting from menopause and testosterone deficiency syndrome [51,52]. To enhance our understanding of the skeletal effects of TDF, investigating castrated animal models is recommended. The existing literature explores bone regulatory mechanisms, such as the NFκB and Wnt signalling pathways [25]. TDF is also known to induce oxidative stress in patients, a phenomenon detrimental to bone health [53,54]. Exploring the involvement of nuclear factor erythroid 2-related factor 2, a major regulator of antioxidant defence mechanisms [55], and assessing mitochondrial redox status in bone cells exposed to TDF could provide valuable insights. Additionally, HIV or hepatitis B infection could cause bone loss independently of treatment [56,57]. Therefore, the bone loss caused by TDF could be compounded by the infections and should be examined with proper study design to delineate the effects between the drug and the infection. 

Despite the establishment of TDF-induced bone loss models in the literature, limitations exist. This review primarily utilised two major databases, potentially overlooking relevant literature, and only included English articles, introducing a language bias. The preponderance of preclinical studies is due to the invasive nature of procedures required for assessing bone quality indices. While advances in skeletal assessment methods like µCT and bone indentation techniques exist, their application in studying the effects of TDF on human bone health is limited. Nevertheless, this scoping review offers a distinctive overview of TDF’s impact on bone quality, a critical component of bone strength beyond BMD.

## 4. Methodology

This scoping review was formulated utilising Arksey and O’Malley’s framework [58] and in compliance with the Preferred Reporting Items for Systematic Reviews and Meta-Analyses extension for scoping reviews (Appendix A) [59]. The following steps were adopted: (1) identifying the research question; (2) identifying the relevant studies; (3) study selection; (4) charting the data; (5) collating, summarising and reporting the results.

### 4.1. Identifying the Research Question

The research question was: what are the effects of TDF on bone quality? Bone quality is defined as the structural and material properties of bone that contribute to its overall strength, durability and ability to function. Its determinants include bone density, microarchitecture, mineralisation and the composition of the bone matrix [60]. This review looked into components other than bone density because this aspect has been reviewed extensively elsewhere. 

### 4.2. Identifying Relevant Studies

A literature search was performed on electronic databases (PubMed and Scopus) in October 2023 using the following search string: tenofovir AND (osteoblast OR osteoclast OR osteocyte OR osteoporosis OR bone). The inclusion criteria for papers were as follows: primary studies involving in vitro and animal models or humans and investigating the effects of TDF on bone quality. Articles without primary results, such as reviews, perspectives, commentary, letters to the editor, books and book chapters, were not considered. Conference abstracts and proceedings were not included due to incomplete data and possible overlap with research articles. Articles not written in English were excluded. No additional filters were applied during the search.

### 4.3. Study Selection

Endnote (version 20.4, Clarivate, London, UK) was used to organise the literature. The search results from the three electronic bases were downloaded. Duplicated items were removed automatically using Endnote and checked manually. The titles and abstracts were screened by K.-Y.C. and T.S.J.S. The inclusion and exclusion criteria were then applied to obtain the full texts of the selected articles. The reference list of the included articles was screened to identify literature that was missed during the search. Any discrepancies were resolved by discussion, and opinions were sought from the other authors. 

### 4.4. Charting the Data

Relevant information from selected studies were extracted by T.S.H.S. and T.S.J.S., which included researchers, publication years, study design (subjects or disease models used, dosage, treatment period) and major findings using a standard Excel table.

### 4.5. Collating, Summarising and Reporting the Results

Instead of synthesising any variables, the scoping review approach was selected due to the heterogeneity of the studies involved and the variables of interest reported. The aim of a scoping review is mainly to provide an overview of the current understanding of the matter. In this regard, the study types, disease models, TDF (dose and treatment period) and major outcomes were summarised and reported. The role of TDF in damaging bone and the research gaps identified are discussed.

## 5. Conclusions

In conclusion, the selected literature suggests that TDF may exert multifaceted effects on bone remodelling, thus altering various bone quality aspects beyond BMD. While rodent studies indicated potential implications for bone histomorphometric indices and biomechanical strength, the findings from zebrafish and monkey models emphasised the complexity of TDF’s influence on bone remodelling and microdamage. Human studies among ART-naive individuals with HIV revealed significant changes in BMD, remodelling markers and cytokine levels, showcasing the clinical relevance of TDF-induced skeletal alterations. Of note, the temporal effects of ART containing TDF, as evidenced by the varying outcomes in fracture healing observed in rat models over different time points, highlight the need for longitudinal assessments in understanding the evolving impact of TDF on bone health.

The evidence summarised underscores the importance of a context-dependent consideration of TDF’s negative skeletal effects. Further research is warranted to elucidate the underlying molecular mechanisms, identify potential mitigating factors and inform clinical strategies for managing bone health in individuals undergoing TDF-based antiretroviral therapy.

## Figures and Tables

**Figure 1 pharmaceuticals-17-00146-f001:**
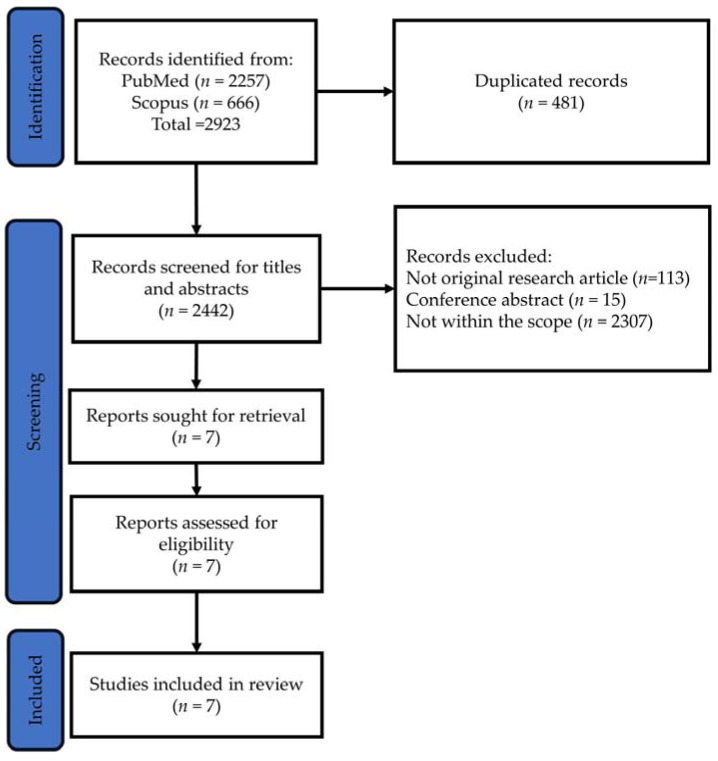
PRISMA flow chart for article search, screening and identification.

**Figure 2 pharmaceuticals-17-00146-f002:**
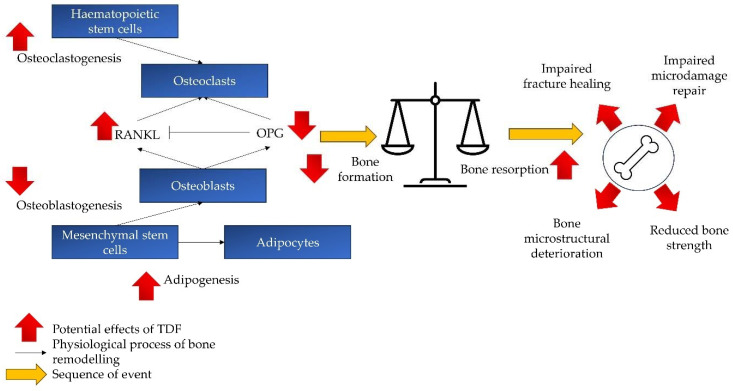
The effects of TDF on bone remodelling and quality. TDF skews the bone remodelling towards the catabolic process (bone resorption > bone formation), thereby compromising bone microstructure and strength. It also impairs bone fracture healing and microdamage repair. Abbreviations: RANKL: receptor activator of nuclear factor kappa beta; OPG: osteoprotegerin; TDF: tenofovir disoproxil fumarate.

**Table 1 pharmaceuticals-17-00146-t001:** Summary of findings from selected studies.

Researchers	Study Design	Findings
Carnovali et al. (2016) [24]	Animals: 3-, 6- and 12-month-old *Danio rerio* AB male zebrafish of similar weight and length Study design: TDF in various concentrations (50 nM, 100 nM and 1 mM)	Osteogenesis in embryos: No significant effect on mineralisation rate of embryos. No significant skeletal or growth deformity or vitality impairment of embryos. Remodelling of scale: ↓ ALP activity in 6-month-old fish exposed to TDF (50 nM, 100 nM and 1 mM). At 100 nM, the reduction in ALP became more significant with age (12 > 6 > 3 months old). ↑ TRAP activity at 100 nM TDF exposure. At 100 nM, the increase in ALP became more significant with age (12 > 6 months old). Phenotype of scales: TDF completely blocked growing circle formation of scale in 6- and 12-month-old fish. TDF ↑ resorption area along the scale’s edge.
Conesa-Buendía et al. (2019) [25]	In vitro study: Bone marrow cells from female C57BL/6 mice were exposed to TDF (10 mM to 1 nM) in the presence of RANKL (osteoclast assays) or osteogenic medium (osteoblast assays) In vivo study: Animals: Male C57BL/6 mice and female C57BL/6 mice Study design: Male C57BL/6 mice: *n* = 10/group Control: saline TDF: 75 mg/kg/day s.c. Dipyridamole: 25 mg/kg/day i.p. Combination: TDF 75 mg/kg/day s.c. + dipyridamole 25 mg/kg/day i.p. Period: 4 weeks Female C57BI/6 mice: *n* = 10/group Sham: not OVX Control: OVX + saline TDF: OVX + 75 mg/kg/day s.c. Dipyridamole: OVX + 25 mg/kg/day i.p. Combination: OVX + TDF 75 mg/kg/day s.c. + dipyridamole 25 mg/kg/day i.p. Period: 5 weeks	In vitro study: Tenofovir ↑ RANKL-induced TRAP-positive cell formation. Tenofovir ↑ cathepsin K, NFATc1 and osteopontin mRNA expression. Knocking down pannexin-1 or connexin-43 ablated the effects of tenofovir on osteoclast and resorption pit formation. A2AR silencing did not reduce the effects of tenofovir on pit formation. Tenofovir activated ERK1/2, p38 and NFκB nuclear translocation during osteoclast differentiation. Tenofovir (IC_50_: 0.4 mM) ↓ osteoblast differentiation in a dose-dependent manner Tenofovir ↑ RANKL mRNA expression and ↓ OPG mRNA expression Tenofovir ↓ nuclear translocation of beta-catenin In vivo study: Whole-body BMD and BMC: ↓ in tenofovir vs. normal control Bone dynamic histomorphometry: ↓ mineral apposition in tenofovir vs. normal control Bone static histomorphometry: ↑ osteoclast number and ↓ osteoblast number in tenofovir vs. normal control µCT: Cortical bone: ↓ BV/TV Trabecular bone: ↓ BV/TV, Tb.N and ↑ Tb.Sp in tenofovir vs. normal control IHC: ↑ TRAP-positive osteoclasts ↑ cathepsin K ↑ macrophages (CD68 + cells) ↑ RANKL-positive cells ↓ OPG-positive cells ↑ sclerostin-expressing osteocytes in tenofovir vs. normal control ALP-positive cells, collagens or I and III: no change with tenofovir Changes in OVX mice were parallel with male mice.
Conradie et al. (2017) [26]	Animals: 12- to 14-week-old male Wistar rats Treatment (10 rats/group): p.o. daily Normal control 1.5 mL water/day LPV/r 70.8 mg/kg/day Stavudine 6.2 mg/kg/day TDF 26.6 mg/kg/day Period: 9 weeks	BMD: no significant difference between TDF and control at the lumbar and femur (average of left and right). Bone mechanical strength: bending stress at max deflection of TDF ↓ vs. control. No other significant difference. Bone dynamic histomorphometry: MS/BS ↓ in TDF and stavudine vs. normal control. BFR/BS ↓ marginally (*p* = 0.06) in TDF vs. normal control. Bone static histomorphometry: ES/BS, Oc.S/BS, N.Oc/TA/mm^2^: no significant increase in TDF vs. normal control. Stavudine was significant vs. normal control. Ob.S/BS and OS/BS: no significant reduction in TDF vs. normal control. Stavudine was significant vs. normal control. Bone marrow adiposity: Number of lipid droplets ↑ in TDF and stavudine vs. normal control.
Matuszewska et al. (2020) [27]	Animals: 8-week-old male albino Wistar rats Treatment (*n* = 12/group): p.o. daily Control group: saline solution (4 mL/kg) EF group: 25 mg/kg of efavirenz and T group: 15 mg/kg of tenofovir disoproxil. Period: 24 weeks	Bone macrometric measurements: Femoral indices, femoral weight and mid-femoral diameter ↓ in TDF vs. normal control. The mid-tibial diameter ↓ in TDF vs. normal control. BMD: total BMD ↓ in TDF vs. normal control after 24 weeks but not 12 weeks. No significant changes in lumbar, tibial and femoral BMD. IHC: ALP and TRAP expression at lumbar 2 did not change in TDF vs. normal control. Bone structural histomorphometry: Femoral TbN ↓ and Tb.Sp ↑ in TDF vs. normal control. No significant changes in other structural indices at femur, tibia and L2. Bone remodelling markers: Serum IGF-1, osteocalcin, TRAP, CTX, ALP, vitamin D, calcium and phosphate—no significant change between TDF and normal control. Bone mechanical strength: Young’s modulus ↓ in TDF vs. normal control. No change in flexural strength and rigidity.
Graham et al. (2022) [29]	Animals: Adult male Wistar rats (450 to 550 g) Study design: *n* = 8/group Group 1: daily oral ART therapy (TDF 30 mg + lamivudine 30 mg + efavirenz 60 mg) Group 2: control Bone fracture: 3 weeks after therapy On one tibial shaft under anaesthesia and fixed with intramedullary nailing Period: 3 weeks pre-fracture + 8 weeks post-fracture	Fracture healing: At week 4, union rate ↓ in the ART group vs. control group. At week 8, no significant difference µCT: At week 8, some fractures of the ART group showed non-union with a gap with two separated fracture ends. Histological fracture: The gap at the fracture ends filled with fibrous tissue. No woven bone or cartilaginous callus in the inter-fragmentary area. Biomechanical strength: No significant difference between ART and normal control. However, biomechanical strength of contralateral tibia ↑ marginally in ART vs. normal control.
Castillo et al. (2002) [28]	Animals: Growing rhesus monkeys (Macaca mulatta) Study design: Control (untreated/uninfected, *n* = 4) TDF-treated/uninfected (*n* = 4) Untreated/SIV-infected (*n* = 12) TDF-treated/SIV-infected (*n* = 13) Dose of TDF: 30 mg/kg; prenatal by transplacental transfer and postnatal	Tenofovir ↑ tibial osteoid seam (marker of bone microdamage) width regardless of SIV infection. SIV ↑ resorption cavity density regardless of tenofovir treatment.
Ramalho et al. (2019) [30]	Patients: 26 ART-naive males with human immunodeficiency virus aged 18 to 40 years. Treatment: TDF + lamivudine + efavirenz	↓ BMD at total hip, femoral neck and lumbar spine post-ART. ↑ osteocalcin and RANKL post-ART. No significant changes in CTX, P1NP, sclerostin or OPG. ↑ intact parathyroid and vitamin D, ↓ FGF-23 post-ART. ↓ TNFα post-ART. IL-6 ↓ but not significant. ↑ cortical thickness post-ART. ↓ cortical porosity but not significant. ↑ OV/BV and Ob.S/BS post-ART. ↑ OcS/BS post-ART. No significant changes in bone dynamic parameters (MS/BS, MAR, BFR, mineralisation lag time). No significant changes in bone protein expression of TNFα, IL-6, IL-1β, RANKL, OPG, FGF-23, sclerostin. IHC: ↑ OPG+ osteoblast lining cells.

Notes: ↑: increase; ↓: decrease. Abbreviations: A2AR: adenosine A2A receptor; ALP: alkaline phosphatase; ART: antiretroviral therapy; BFR/BS: bone formation rate/bone surface; BMD: bone mineral density; BV/TV: bone volume/total volume; CTX: C-terminal telopeptide; ERK: extracellular-signal-regulated kinase; ES/BS: eroded surface/bone surface; FGF23; fibroblast growth factor 23; IGF-1: insulin-like growth factor-1; IHC: immunohistochemistry; IL: interleukin; Lpv/r: lopinavir / ritonavir; MAR: mineral apposition rate; MS/BS: mineralising surface/bone surface; NFATc1; nuclear factor of activated T cells, cytoplasmic, calcineurin-dependent 1; NFκB: nuclear factor kappa beta; N.Oc/TA: osteoclast number/total area; Ob.S/BS: osteoblast surface/bone surface; Oc.S/BS: osteoclast surface/bone surface; OPG: osteoprotegerin; OS/BS: osteoid surface/bone surface; OV/BV: osteoid volume/bone volume; OVX: ovariectomised; P1NP: procollagen-type I N-propeptide; RANKL: receptor activator of nuclear factor kappa beta; SIV: simian immunodeficiency virus; Tb.N: trabecular number; Tb.Sp: trabecular separation; TDF: tenofovir disoproxil fumarate; TNFα: tumour necrosis factor-alpha; TRAP: tartrate-resistant acid phosphatase; µCT: micro-computed tomography.

## Data Availability

Data sharing is not applicable.

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
