# Peer review of "Effects of Tenofovir Disoproxil Fumarate on Bone Quality beyond Bone Density—A Scoping Review of the Literature"

_pharmaceuticals, 2024, doi:10.3390/ph17020146_

Round 1

Reviewer 1 Report

Comments and Suggestions for Authors

The scoping review by Singh and coworkers aims at shedding light on the effects of tenofovir on bone microarchitecture, healing potential and strength.

Within the literature search performed to prepare this review, only 7 research articles were selected in the end. Thus, the driven conclusions could be a bit limited, although the need to start focusing on how TDF affects bone quality across species is urgent. In addition, the 7 selected articles refer to different animal models and observed TDF effects for different times. However, these limitations were clearly stated in the discussion section.

The review was designed in a rigorous and systematic way and is written in good English. As a minor suggestion, I believe that lines 186-196 and lines 213-232 should be shortened or removed because, in this reviewer opinion, they present basic information unsuitable for a specialized-journal review. A graphical abstract describing TDF impact on bone quality should be added to attract more readers.

Author Response

Dear reviewer,

We appreciate the throughout examination of our manuscript and the constructive comments provided by you. We have replied to each of the comments in a point-to-point manner in the attached response sheet. Thank you.

Reviewer 2 Report

Comments and Suggestions for Authors

Dear Authors,

After reviewing the manuscript, I have identified specific areas that require corrections and explanations.

Major points

Tenofovir is used not only as a part of ART treatment of HIV infection. The WHO indicates that chronic hepatitis B infection should be treated with tenofovir (or entecavir). This piece of information is vital for this review since it greatly increases the number of patients potentially affected by the side effects of chronic treatment with tenofovir, including the influence on bone quality. You should have included papers on the influence of tenofovir on bone mineral density among patients with chronic hepatitis B, e.g. papers by Gill et al. (Gill et al., 2014) or Kahraman et al. (Kahraman et al., 2022).

Moreover, some data (Dessordi et al., 2021) suggest that high resorption activity of bone tissue in hepatitis B virus-infected patients may occur independently of the use of antiretrovirals. Such activity should also be taken into account.

Interestingly, the study by Da Wang et al. (Da Wang et al., 2023) showed that switching antiviral regimens for chronic hepatitis B from tenofovir disoproxil fumarate to tenofovir alafenamide or entecavir leads to statistically significant improvement in bone mineral density. This would also be valuable information in the presented review. Although you mentioned (lines 65-67) that “individuals exposed only to TDF exhibited a more prominent loss in BMD as compared to those who were exposed to other antiretroviral drugs or a multi-drug regimen”, the information that changing the pharmaceutical form of tenofovir, from disoproxil fumarate to alafenamide may be beneficial, is an interesting observation.

Minor points

Lines 8-9: “Tenofovir disoproxil fumarate (TDF) is a widely used pharmacological agent for the treatment of human immunodeficiency virus.” HIV is a virus, not a disease; please change, e.g. “for the treatment of HIV infection.”

Please write in vitro and in vivo in italics.

References

Gill, U. S., Zissimopoulos, A., Al-Shamma, S., Burke, K., McPhail, M. J., Barr, D. A., Kallis, Y. N., Marley, R. T., Kooner, P., Foster, G. R., & Kennedy, P. T. (2015). Assessment of bone mineral density in tenofovir-treated patients with chronic hepatitis B: can the fracture risk assessment tool identify those at greatest risk? The Journal of Infectious Diseases, 211(3), 374–382. https://doi.org/10.1093/infdis/jiu471

Kahraman R, Şahin A, Öztürk O, Çalhan T, Sayar S, Kanat E, Doğanay L, Özdil K. Effects of Long-Term Tenofovir and Entecavir Treatment on Bone Mineral Density in Patients with Chronic Hepatitis B. Turk J Gastroenterol. 2022 Jan;33(1):35-43. doi: 10.5152/tjg.2020.18024. PMID: 35040786; PMCID: PMC9128468.

Dessordi, R., Watanabe, L.M., Guimarães, M.P. et al. Bone loss in hepatitis B virus-infected patients can be associated with greater osteoclastic activity independently of the retroviral use. Sci Rep 11, 10162 (2021). https://doi.org/10.1038/s41598-021-89486-9

Da Wang, F., Zhou, J., Li, L. Q., Li, Y. J., Wang, M. L., Tao, Y. C., Zhang, D. M., Wang, Y. H., & Chen, E. Q. (2023). Improved bone and renal safety in younger tenofovir disoproxil fumarate experienced chronic hepatitis B patients after switching to tenofovir alafenamide or entecavir. Annals of hepatology, 28(5), 101119. https://doi.org/10.1016/j.aohep.2023.101119

Comments on the Quality of English Language

The quality of the English language seems adequate, with only a few minor corrections needed.

Author Response

(The authors gave the same response as above.)

Round 2

Reviewer 2 Report

Comments and Suggestions for Authors

Dear Authors,

I appreciate your response to my previous comments. I wanted to inform you that the manuscript has been properly corrected.

Comments on the Quality of English Language

The English language quality is sufficient with minimal corrections required.